# Aliens Coming by Ships: Distribution and Origins of the Ocellated Skink Populations in Peninsular Italy

**DOI:** 10.3390/ani12131709

**Published:** 2022-07-01

**Authors:** Emiliano Mori, Franco Andreone, Andrea Viviano, Francesco Paolo Faraone, Matteo Riccardo Di Nicola, Bernardo Borri, Giacomo Bruni, Giuseppe Mazza, Riccardo Banchi, Marco Zaccaroni, Sergio Mezzadri, Mariella Baratti

**Affiliations:** 1Institute of Terrestrial Ecosystems IRET-CNR, Via Madonna del Piano, 10, 50019 Sesto Fiorentino, Italy; emiliano.mori@cnr.it (E.M.); a.viviano@studenti.unipi.it (A.V.); 2Museo Regionale di Scienze Naturali, Via Giovanni Giolitti, 36, 10123 Torino, Italy; 3Independent Researcher, Viale Regione Siciliana S. E., 90129 Palermo, Italy; paolofaraone@libero.it; 4Unit of Dermatology, IRCCS San Raffaele Hospital, Via Olgettina, 60, 20132 Milan, Italy; matteodinicola86@libero.it; 5Independent Researcher, Via dei Massoni, 4, 50019 Sesto Fiorentino, Italy; borri.pachypus@gmail.com; 6Societas Herpetologica Italica, Commissione Atlante, Viale Palmiro Togliatti, 50019 Sesto Fiorentino, Italy; giacomobruni90@gmail.com; 7CREA-Research Centre for Plant Protection and Certification (CREA-DC), Via di Lanciola, 12/A, Cascine del Riccio, 50125 Florence, Italy; giuseppe.mazza@crea.gov.it; 8Independent Researcher, Via Anita Garibaldi, 73, 57025 Piombino, Italy; riccardo.banchi@gmail.com; 9Department of Biology, University of Firenze, Via Madonna del Piano, 6, 50019 Sesto Fiorentino, Italy; marco.zaccaroni@unifi.it; 10Independent Researcher, Via Palmerio, 27/c, 29121 Piacenza, Italy; sergio.mezzadri@libero.it; 11Institute of Biosciences and Bioresources IBBR-CNR, Via Madonna del Piano, 10, 50019 Sesto Fiorentino, Italy; mariella.baratti@cnr.it

**Keywords:** *Chalcides ocellatus*, mitochondrial DNA, port areas, species introduction, Reptilia

## Abstract

**Simple Summary:**

Commercial routes are reported as the main cause of biological invasions. Particularly, naval trade may accidentally bring several species to new areas where they are not native. This is particularly evident for coastal areas, where most biological invasions occur. In our work, we reported, for the first time, the presence of the ocellated skink, native to the largest Italian islands (Sardinia, Sicily and surrounding islets in a port area of continental Central Italy). We collected several individuals of this alien population and we sampled them for molecular analyses, comparing them with those naturally occurring in Sardinia, Sicily and the Mediterranean basin, including individuals accidentally introduced to peninsular Southern Italy. Differently from what previously suggested, the nucleus in Portici (Southern Italy) may have originated from Sardinia. The intense cork trade and touristic traffic between Sardinia and Southern Tuscany may have been responsible for the introduction of this lizard also to Central Italy.

**Abstract:**

The ocellated skink (*Chalcides ocellatus*) is a widespread lizard, naturally distributed between the Maghreb and coastal Pakistan, with few insular populations in the Mediterranean coastal area. Some populations of this species have also been recorded in peninsular Italy, Campania and Southern Tuscany due to accidental introductions via touristic and commercial routes. In this work, we conducted genetic analyses on mitochondrial DNA COXI, cytb and 16S mtDNA genes on a sample of Italian insular and peninsular populations. Differently from what previously suggested, the nucleus in Portici (Southern Italy) may have originated from Sardinia. The intense trade and touristic traffic between Sardinia and Southern Tuscany may have been responsible for the introduction of this lizard also to Central Italy.

## 1. Introduction

Maritime transport supports 80% of the current global trade, a quarter of which crosses the Mediterranean [1,2,3]. The increasingly heavy commercial traffic has required considerable human movements and trade in goods, which have in turn promoted and encouraged the alteration of the native distribution of many animal and plant species [1,2,4]. Ports and naval trade represent one of the main pathways of introduction and hubs of alien and invasive species [2,3,5,6,7], being a strategic gate for access to the main tourist areas and routes across coastal areas. In this context, over 600 municipalities in Italy locate along the coastline, hosting 27% of the human population of Italy and constituting a nerve center for both tourism and trade. Furthermore, Italy also holds the European record for the number of alien species, including 30% of its terrestrial and freshwater vertebrates [8]. Among those, the detection of alien reptiles has increased in recent decades in coastal areas following the increased importance of reptiles as pets and, mostly, accidental introductions with the brick, plant, and lumber trade [9,10]. Although most releases fail to establish a reproductive population, as regarding only single individuals, many reptile species have successfully colonized areas where there were historically absent. Human-mediated dispersal has allowed several gecko species to reach most Italian regions even far from the coastline (e.g., *Hemidactylus turcicus*, *Mediodactylus kotschyi*, and *Tarentola mauritanica*) [11,12,13,14]). Two populations of Mediterranean chameleon (*Chamaeleo chamaeleon*) occur in Southern Italy, due to multiple introduction events [15,16]. Similarly, the Brahminy blindsnake *Indotyphlops braminus* is a semi-fossorial snake native to the Indo-Malayan region, able to survive also inside pots [17]. This adaptability has allowed it to disperse outside its native range through the international plant trade, with at least two populations also in Italy, one in Sicily and one on Ischia island, Campania [17,18].

The ocellated skink (*Chalcides ocellatus*) is a widespread lizard species naturally occurring throughout Northern Africa, Western Asia, and South-Eastern Europe [19,20]. Traditionally, the subspecies *C. o. tiligugu* is distributed throughout Italy, i.e., in Sicily and Sardinia [10] and Tunisia [10,21,22]. Two other subspecies from Linosa (*C. o. linosae*) and Lampedusa islands (*C. o. zavattarii*) are not genetically supported [23] and, thus, they should be considered as invalid [10]. A small population is also present in the surroundings of Portici (province of Naples), most likely as the result of an accidental introduction [24]. However, no information is available to confirm its actual origin. In spring 2021, a photo showing two individuals of ocellated skink in Tuscany, Central Italy, was posted on Facebook (Figure 1A), and solicited our attention to the origin of these animals. Other photos followed and confirmed the species identification (Figure 1B,C).

Previous molecular phylogenetic inferences [19,20,21] include south-eastern Mediterranean and northern African populations, with a few samples from the two Italian major islands, Sicily and Sardinia. The recent findings of some samples from Peninsular Italy (Piombino, Livorno and Portici, Naples) allowed us to investigate their genetic relationships and to infer their origin.

## 2. Materials and Methods

### 2.1. Sample Collection

In Tuscany, samples were collected in a crop area with small vegetable gardens in the Piombino municipality, i.e., in the immediate surroundings of the port and in Torre Mozza (Environment Ministry MATTM permits: 0065971, 18 June 2021; ISPRA: Prot. 31076, 11 June 2021: Figure 2).

Skinks were searched in June to September 2021, following the monitoring program [25]. Four pitfall traps baited with insect larvae were also employed. They were checked twice a day to verify the presence of any animal fallen inside. Moreover, we also provided a shelter inside and a small container with water to assure survivorship. Each individual was sampled by cutting a small piece (1–2 mm) of the tail. This action does not affect skink health, since this species, like others of the same family, is particularly prone to autotomy, i.e., voluntary tail-loss allowing escaping for defense. Tissue samples were preserved in absolute ethanol before genetic analyses. The sample from Portici was derived from an individual found dead on a sidewalk. Further samples were previously collected in other localities included in the skink range by some authors (Sicily and Sicilian Islands: Giuseppe Mazza, Matteo Riccardo Di Nicola and Francesco Paolo Faraone; Sardinia: Matteo Riccardo Di Nicola and Sergio Mezzadri), and stored in absolute ethanol in museums (sample MZUT R203, Linosa) or private collections (Table 1, Figure 2).

### 2.2. Molecular Analyses

Samples of *C. ocellatus* collected in Italy were sequenced and analyzed, whereas other sequences of congeneric species (*C. chalcides* and *C. viridanus*) were employed as outgroups (Table 1). All samples were preserved in 96% ethanol and genomic DNA was extracted using QIAGEN Blood and Tissue kit (Qiagen Inc., Germantown, MD, USA), following the manufacturer’s protocol. Mitochondrial DNA sequencing PCR amplification products were obtained from three genes: the Cytochrome Oxidase I (hereafter, COXI), 12S mitochondrial gene (hereafter, 12S) and Cytochrome b (hereafter, cytb). The primers used are described in Table 2.

The PCR profiles were the same reported by Kornilios et al. [19] for 12S and cytb; for COXI, we followed the protocol described by Baratti et al. [30]. PCR products were run on a 1.5% agarose gel purified (ExoSAP-IT, Amersham Biosciences, Amersham, UK) and sequenced with a sequencing kit (ABI Big Dye Terminator Cycle Sequencing v. 2.0- ABI PRISM, Applied Biosystems, Foster City, CA, USA). GenBank accession numbers are reported in Table 1. Electropherograms were visualized with CHROMAS 1.45 (http://www.technelysium.com.au (accessed on 9 June 2022)). The sequences were manually corrected and then aligned using CLUSTALX 1.81 [31]. In order to determine if sequences were nuclear (NUMTs [32,33]) or mitochondrial copies, we followed three steps. First, sequence chromatograms were checked for double signals. Next, coding sequence alignments were inspected for frameshift mutations and/or stop codons. Finally, the corrected sequences were compared to those in GenBank: we compared our sequences to the ones deposited in the NCBI database using BLASTx and BLASTn. We analyzed two datasets: one with sequences by cytb (Dataset 2), only because it was the gene with more sequences in GenBank and it allowed us to compare our sequences with a higher number of sequences (Table 1); all three genes constituted the other dataset (Dataset 1).

For both datasets, we carried out a phylogenetic reconstruction by Neighbor-Joining (NJ), Bayesian (BI) and a Maximum Likelihood (ML) phylogenetic analysis. Analyses were carried out using genetic models selected by jModelTest [34], with the Akaike Information Criterion (AIC). The HKY (Hasegawa–Kishino–Yano) nucleotide substitution model was selected for cytb, whilst GTR (General Time Reversible) was used for the combined dataset. Models were corrected for rate heterogeneity among sites with a Gamma (G) distribution [35]. The NJ was performed by MEGA software and 1000 bootstrap replicates. The BI analysis was conducted with MrBayes v.3.12 [36], using the best model selected by Modeltest. Four chains of Markov chain Monte Carlo (MCMC) were run simultaneously and sampled every 1000 generations for 4 million generations. The first 1000 sampled trees from each run were discarded as burn-in. The proportion of trees that contained the clade was given as the posterior probability (PP) to estimate the robustness of each clade. Branch supports were assessed by 100 non-parametric bootstrap replicates. A strict consensus tree was calculated when there was more than one tree. ML phylogenetic analysis was conducted through the SeaView software [37]. We selected the optimized choices and we obtained the tree-searching operations by Nearest-Neighbor Interchange (NNI) and Subtree Pruning–Regrafting (SPR).

## 3. Results

In our work, we confirmed, for the first time, the presence of a population of ocellated skink on the Tuscan coast, between the ports of Piombino and Torre Mozza. The population was confirmed to be reproductive, as two juveniles were also observed. In this area, seven adult individuals of ocellated skink (*N* = 4 males, 3 females) were sampled and immediately released.

After alignment and trimming, we examined 585 base pairs for COXI, 366 base pairs for 12S and 287 base pairs for cytb. The sequences were analyzed in two datasets: the former (Figure 3), named Dataset 1, included all three genes (1238 bp), including the first COXI barcode sequences for this species, and the latter, named Dataset 2, included a higher number of sequences, but only the cytb gene portion (Figure 4).

The different phylogenetic reconstructions applied to both datasets gave similar results (Figure 3 and Figure 4). Within *C. ocellatus* sequences, two main clades occurred. The former included southern Mediterranean populations (Greece, Syria, Egypt, Algeria, Morocco and Saudi Arabia), except for Tunisia, which resulted as the sister clade of all the other *C. ocellatus* groups. The second clade was constituted by the group of Tunisian and Italian populations. However, the last group looked quite close to Sardinian populations but not to the Sicilian or other peninsular populations, even though the tree is largely not solved at the interspecies level. The Linosa sample was included in the Sicilian clade, even if the relationships among Sicilian populations were not highly supported. However, the phylogenetic relationships within different Italian populations were not well resolved. The Dataset 1 tree showed two sister groups: populations from Tuscany (Piombino and Torre Mozza, Livorno) in one group and Sicilian, Sardinian and Portici populations in the other group. Sicilian samples are grouped together and they have the Pantelleria sample as a sister group. The Portici sample looked quite close to Sardinia, whereas Linosa confirmed its position inside the Sicilian group even if the support values at nodes of this group were not high.

Sardinian populations were very close to one of the Tunisian sequences, and appeared as the only one with such high genetic affinity, as enlightened also by genetic distances (Table 3).

## 4. Discussion

The ocellated skink is a widespread lizard species, with populations throughout the Mediterranean basin, up to the Middle East. The species may have evolved in Northern Africa and, afterward, expanded its range until coastal Pakistan [19]. In our work, we reported the presence of this species in a coastal area of Central Italy (Tuscany) for the first time, and we reconfirmed the presence of this species in the surroundings of Naples (Southern Italy). We also performed the first molecular analyses on both peninsular populations. The occurrence of the ocellated skink in Mediterranean islands and the lack of its presence in Corsica and the European mainland strongly suggest recent introduction events, i.e., after the Last Glacial Maximum (i.e., 18,000 years ago [38]), as is currently assumed for other Sardinian reptiles such as *Testudo graeca* [39], *Chalcides chalcides* [21], *Natrix maura* [40] and *Hemorrhois hippocrepis* [41]. Particularly, ocellated skinks may have reached Mediterranean islands through commercial routes between Europe and North Africa [19]. Carranza et al. [21] sustained that almost all the populations of *C. ocellatus* clade can be assigned to a single species, although they exhibit intraspecific genetic divergence with 6–8% values in cytb + 12S. Consequently, a split in different taxa must be evaluated.

We are aware of the limitation due to our small sample size, as our analysis was based on few mitochondrial DNA-only loci from a limited number of individual samples. However, we provided the first COXI barcoding sequences for this species and the first available information on the origins of the only two populations occurring in peninsular Italy; the one in Piombino is still unreported in the scientific literature.

In our analyses, the Italian populations showed very low genetic distance values, with the highest between the Tuscany coast (Torre Mozza and Piombino) and the other populations (Table 3). Sicilian and Sardinian samples always showed low values for both the COXI and 12S + cytb divergences. Maio et al. [24] suggested that the small population present in the surroundings of Portici (province of Naples) is most likely the result of an accidental introduction from Sicily. Conversely, molecular analyses suggested that it may actually represent an introduction from Sardinia. The first record of the ocellated skink in Campania dates back to 1863 in the former Royal Park of Portici (Naples). However, in this area, the population of this lizard underwent several fluctuations, with no individual intercepted between 1990 and 2014. Afterwards, at least two observations of individuals morphologically ascribed to the Mediterranean subspecies *C. o. tiligugu* were detected (the first one reported by [24], the second one reported in this work), suggesting that the population is not extinct. As for Tuscany, at least seven individuals, morphologically ascribed to *C. o. tiligugu* (i.e., showing light dorso-lateral bands running from the head to the tail root [10]) were detected through addressed surveys in 2021; the presence of juveniles provided support to local reproduction events. The presence of the ocellated skink in the immediate surroundings of the Piombino port area suggests that these individuals have been introduced from Sardinia, possibly via the cork trade between Sardinia and the Italian mainland [42]. This hypothesis is also confirmed by the fact that the vegetable gardens where ocellated skinks were detected in Piombino are located near a parking area for trucks of cork from Sardinia.

Other Sardinian species followed the same colonization pattern, e.g., the Mediterranean snakefly (*Fibla maclachlani*), recorded since 2005 also in the coastal part of Southern Tuscany, where it is currently expanding [43]. This species is endemic to the Mediterranean largest islands and can survive at its larval stages in cork bark [43]. The ocellated skink is quite common in Mediterranean habitats characterized by the presence of *Quercus suber* L. [44]. The intense trade and touristic routes between Sardinia and Piombino may thus have promoted the colonization of the Italian mainland by several Sardinian species [43,45,46]. However, despite representing an actual introduction, we suggest that impacts of *C. ocellatus* on the Italian mainland would not represent a threat to native biodiversity, given the low population densities [24]. If populations were to increase in size, some competition with native lizards may occur [44]. Further monitoring is necessary to determine the future increase in population size and geographic range in introduction areas.

Our molecular phylogeny did not confirm previous hypotheses on the existence of a different subspecies from Linosa island (*C. o. linosae* [10]), which may thus represent a different insular form of the same subspecies occurring in Sicily, thus confirming the conclusions by Stöck et al. [23]. However, sequences shown by Stöck et al. [23] were not retrieved from online genetic databases. In our analyses, the Linosa population was always inside the Sicilian group, even though the low support node values did not suggest an exact position.

## 5. Conclusions

Despite the wide distribution range of the ocellated skink, its scattered presence in Europe makes this species a conservation concern [25]. Thus, although representing introduced populations, the monitoring of ocellated skinks in peninsular Italy deserves future attention also in port areas.

Port areas constitute a typically perturbed anthropogenic ecosystem, widespread and highly globally interconnected. Therefore, they have considerable potential to be hubs for the diffusion of aquatic and terrestrial alien species [3,8]. The stabilization processes of new species are often reported starting from the port or circum-port areas [3]. In this context, our findings of a population of ocellated skink in a Tuscan coastal area and the confirmation of the presence of the species near an important port area in Campania provide further confirmation of the importance of ports as pathways of alien species introduction [3].

## Figures and Tables

**Figure 1 animals-12-01709-f001:**
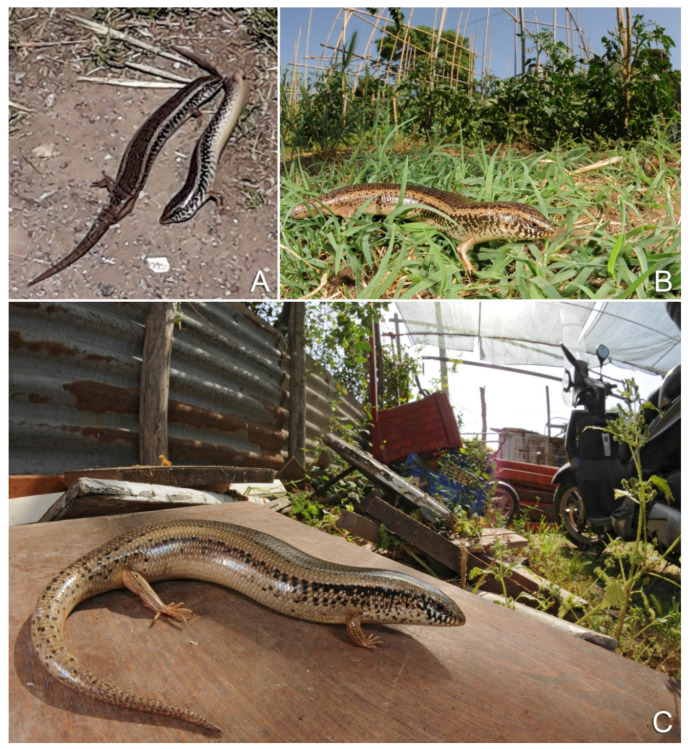
(**A**) the first photograph documenting the ocellated skink in Tuscany (© Ugo Preziosi); (**B**,**C**) individuals of this species found in the surroundings of Piombino port (province of Livorno, Central Italy, © Matteo Riccardo Di Nicola).

**Figure 2 animals-12-01709-f002:**
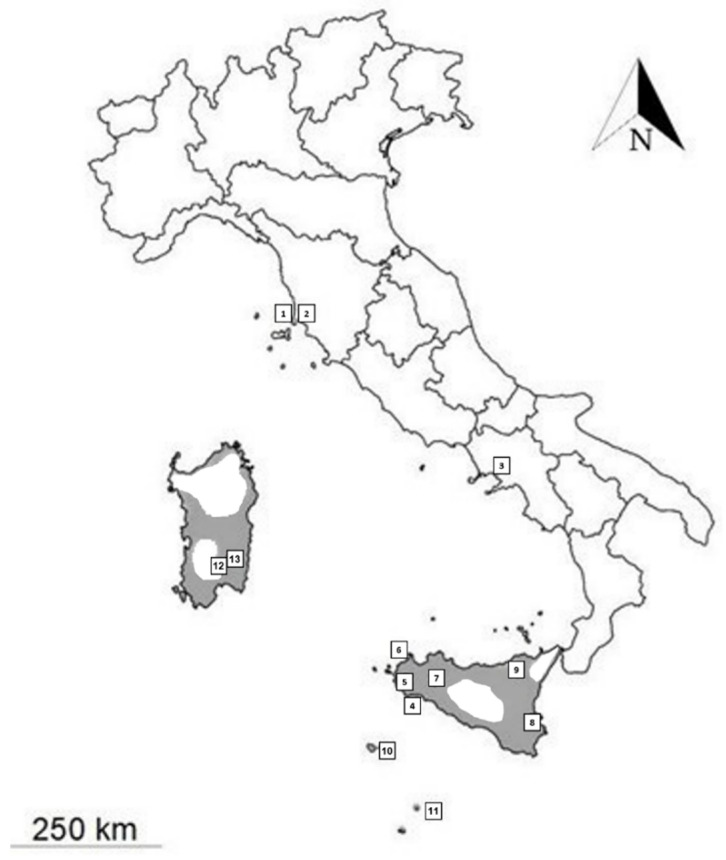
Updated distribution (gray areas) of the ocellated skink in Italy and sites of sample origin (from [10], modified): (1) Piombino; (2) Torre Mozza; (3) Portici; (4) Triscina di Selinunte; (5) Mazara del Vallo; (6) Riserva dello Zingaro; (7) Ficuzza; (8) Siracusa; (9) Rocche del Crasto; (10) Pantelleria island; (11) Linosa Island; (12) Serdiana; (13) Dolianova. One sample was analyzed from each place.

**Figure 3 animals-12-01709-f003:**
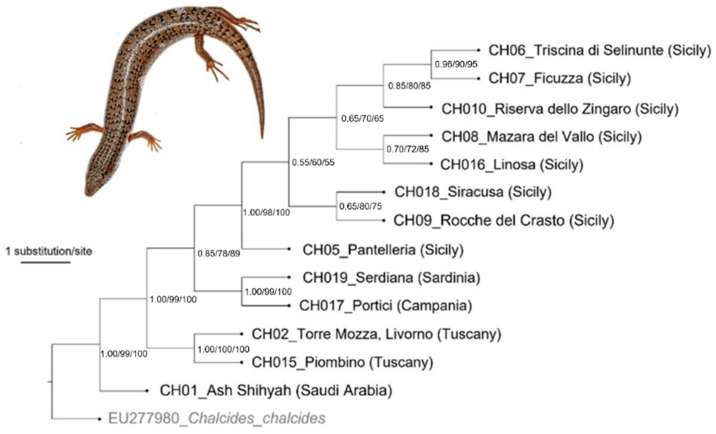
Maximum-likelihood phylogenetic tree of *Chalcides ocellatus* mtDNA sequences (three genes, Dataset 1). The numbers at nodes indicate Bayesian posterior probability, Maximum-Likelihood and Neighbor-Joining bootstrap values.

**Figure 4 animals-12-01709-f004:**
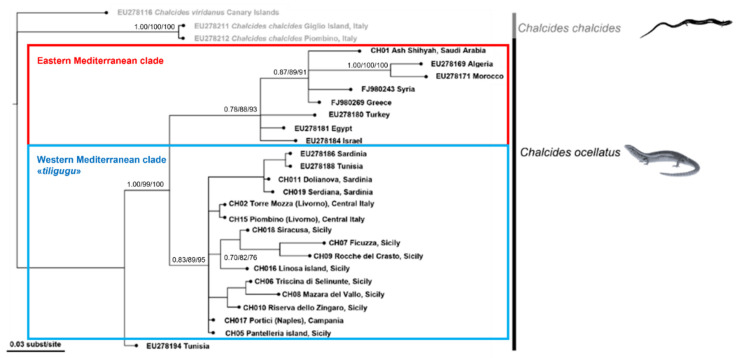
Maximum-likelihood phylogenetic tree of *Chalcides ocellatus* mtDNA sequences (cytb, Dataset 2). The numbers at nodes indicate Bayesian posterior probability, Maximum-Likelihood and Neighbor-Joining bootstrap values. Main clades in the *C. ocellatus* group are shown.

**Table 1 animals-12-01709-t001:** Samples of *Chalcides ocellatus* and outgroups (*C. chalcides*, *C. viridanus*) included in our analyses, locality of collection, coordinates, amplified genes and GenBank accession numbers (COXI, 12S, cytb). *, sample MZUT R203 (Turin, Piedmont, NW Italy).

							Dataset 1		
									Dataset 2
Species	Region/District	Locality (Point in Figure 2)	Latitude	Longitude	Sample Label	Accession Numbers	COXI	12S	CYTB
*C. ocellatus*	Saudi Arabia	Ash Shihyah	26.269	43.597	CH01_ARABIA	ON534012, ON534203, ON551370	X	X	X
*C. ocellatus*	Tuscany/Livorno	Torre Mozza (2)	42.946	10.693	CH02_TORRE_MOZZA	ON534009, ON534204, ON551371	X	X	X
*C. ocellatus*	Sicily/Trapani	Pantelleria island (10)	36.817	12.003	CH05_PANTELLERIA	ON534007, ON534202, ON551381	X	X	X
*C. ocellatus*	Sicily/Trapani	Triscina di Selinunte (4)	37.584	12.785	CH06_SELINUNTE	ON534013, ON534206, ON551377	X	X	X
*C. ocellatus*	Sicily/Trapani	Ficuzza (7)	37.883	13.373	CH07_FICUZZA	ON534014,ON534207, ON551375	X	X	X
*C. ocellatus*	Sicily/Trapani	Mazara del Vallo (5)	37.650	12.599	CH08_MAZARA	ON534015, ON534208, ON551378	X	X	X
*C. ocellatus*	Sicily/Messina	Rocche del Crasto (9)	38.023	14.746	CH09_MESSINA	ON534019, ON53420, ON551376	X	X	X
*C. ocellatus*	Sicily/Trapani	Riserva dello Zingaro (6)	38.081	12.808	CH010_ZINGARO	ON534016, ON534209, ON551379	X	X	X
*C. ocellatus*	Sardegna/Sud Sardegna	Dolianova (13)	39.374	9.180	CH011_SARDINIA	ON551373			X
*C. ocellatus*	Tuscany/Livorno	Piombino (1)	42.924	10.544	CH015_PIOMBINO	ON534010, ON534205, ON551372	X	X	X
*C. ocellatus*	Sicily/Agrigento	Linosa island (11) *	35.870	12.862	CH016_LINOSA	ON534017, ON534210, ON551382	X	X	X
*C. ocellatus*	Campania/Naples	Portici (3)	40.804	14.349	CH017_PORTICI	ON534011, ON534211, ON551380	X	X	X
*C. ocellatus*	Sicily/Siracusa	Siracusa (8)	37.080	15.286	CH018_SIRACUSA	ON534018, ON534212, ON551374	X	X	X
*C. ocellatus*	Sardegna/Sud Sardegna	Serdiana (12)	39.372	9.159	CH019_SARDINIA	ON534008,ON534213, ON551383	X	X	X
*C. ocellatus*	Algeria				EU278169_ALGERIA	EU278169			X
*C. ocellatus*	Morocco				EU278171_MOROCCO	EU278171			X
*C. ocellatus*	Turkey				EU278180_TURKEY	EU278180			X
*C. ocellatus*	Syria				FJ980143_SYRIA	FJ980143			X
*C. ocellatus*	Greece				FJ980268_GREECE	FJ980268			X
*C. ocellatus*	Egypt				EU278181_EGYPT	EU278181			X
*C. ocellatus*	Israel				EU278184_ISRAEL	EU278184			X
*C. ocellatus*	Tunisia				EU278194_TUNISIA	EU278194			X
*C. ocellatus*	Sardinia				EU278194_SARTIL	EU278186			X
*C. ocellatus*	Tunisia				EU278188_TUNTIL	EU278188			X
*C. chalcides*	Italy	Giglio island (Grosseto)			EU278211_CCGIGLIO	EU278211			X
*C. chalcides*	Italy	Piombino (Livorno)			EU278212_CCPIOMBINO	EU278212			X
*C. viridanus*	Spain	Canary Islands			EU278116_CVCANARY	EU278211			X

**Table 2 animals-12-01709-t002:** Primers used for the molecular analyses of *Chalcides ocellatus*.

Target Gene	Label	Sequence 5′-3′	Reference	Fragment Length (bp)
cytb	L14841	AAAAAGCTTCCATCCAACATCTCACATGATGAAA	[26]	325
	H15149	AAACTGCAGCCCCTCAGAATGATATTTGTCCTCA		
12S	12StPhe	AAAGCACRGCACTGAAGATGC	[27]	350
	12 g	TATCGATTATAGGACAGGCTCCTCTA	[28]	
COXI	LCO1490	GGTCAACAAATCATAAAGATATTGG	[29]	670
	HCO 2198	TAAACTTCAGGGTGACCAAAAAATCA		

**Table 3 animals-12-01709-t003:** Genetic distances calculated among the studied populations of *Chalcides ocellatus* with cytb distances above diagonal and 12S + COXI below the diagonal.

	Torre Mozza	Piombino	Serdiana	Siracusa	Ficuzza	Rocche Crasto	Triscina Selinunte	Mazara Vallo	Riserva Zingaro	Portici	Pantelleria	Linosa
Torre Mozza		0.5%	2%	2%	2%	2%	2%	2%	2%	2%	2%	2%
Piombino	1.4%		2%	2%	2%	2%	2%	2%	2%	2%	2%	2%
Serdiana	4.3%	3.6%		0%	0%	0%	0%	0%	0%	2%	0%	0%
Siracusa	2.5%	3.5%	1.4%		0%	0%	0%	0%	0%	0%	0%	0%
Ficuzza	4%	4%	2.6%	1.6%		0%	0%	0%	0%	0%	0%	0%
Rocche Crasto	2.5%	3.2%	1.4%	0.1%	1.6%		0%	0%	0%	0%	0%	0%
Triscina Selinunte	3.2%	2.5%	2.1%	0.1%	0.5%	1%		0%	0%	0%	0%	0%
Mazara Vallo	2.3%	4%	1.2%	0.2%	1.4%	1.4%	2%		0%	0%	0%	0%
Riserva Zingaro	2%	2%	1.7%	0.7%	1%	1%	1%	3%		0%	0%	0%
Portici	4.3%	3.6%	0.2%	1%	2.6%	1.8%	2%	4%	1%		0%	0%
Pantelleria	2.5%	2.5%	1.7%	1%	1.6%	0.7%	2%	0.5%	1%	0%		0%
Linosa	2.5%	2.5%	1.2%	0.2%	0.9%	1.4%	1.4%	1.5%	1.3%	2%	2%	

## Data Availability

All data are included in the MS or deposited on GenBank.

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
