# Peer review of "Aliens Coming by Ships: Distribution and Origins of the Ocellated Skink Populations in Peninsular Italy"

_animals, 2022, doi:10.3390/ani12131709_

Round 1
Reviewer 1 Report
The present genetic study of ocellated skink had results very interesting. Congratulations for your work!
However, it is necessary to improve the manuscript at some points before the publication:
- In general, I recommend an English review, as some sentences are incorrectly structured and some expressions are literally traduced from Italian.
- Please pay attention at the page numbering. It starts 1 to 4 and after table 1, it starts again 1 to 7....
- L32. You repeat twice the verb to sample in the same sentence. Please, use a synonym for one of them. This sentence is hard to read and understand, please reword it.
- L38. Put Chalcides ocellatus in brakets, please.
- L39. Please change thr to the.
- L52. "transport goods"... I think this expression is not well traduced...
- L56. Italy includes more municipalities than only coastal ones. Please reformulate the sentence.
- L62. Please change "If on one hand" to "In spite of" or "Despite".
- L63-64. Please reformulate "mainly because of single released individuals"
- L67. Please, put Mediodactylus kotschyix in brakets.
- L89. "mainly dealt with" doesn't make sense. Please reformulate.
- L99. Please change "cultivated area" to "crop area"
- L111. Please add the year of sample collection, not only the months.
- L121. I don't find in the text the meaning of GM, FPF, MDN and SM. For the first time you use an abbreviation in a scientific text it is important to put the meaning. For example: ... the pulse-fiel gel electrophoresis (PFGE)... Please do the same in L162 and 163 with HKY and GTR respectively.
- L143. "...were the same reported by (19)...". In order to make easier the reading, please change this to "...were the same reported by Kornilios et al. (2010) (19). Please do the same in L144, 262 and 263 with references 30 and 23 respectively.
- L150. "the following steps were followed". Please reformulate...
- Figure 3 and 4. It results very difficult to read because of the small print. Please try to enlarge the text...
- Table 3. The title is incorrect.
- L253. Please put Fibla maclachlani in brackets.
- L272-273. You repeat Therefore twice. Please change one of them...
Author Response
Torino, 27.06.2022
Dear Editorial Board,
We have adapted our MS to all of your suggestions.
All changes in the text are shown in red.
Hopefully, my coauthors’ and my MS can now be accepted.
Sincerely,
Dr. Franco Andreone
Corresponding Author
on behalf of all coauthors
Reviewer 1
The present genetic study of ocellated skink had results very interesting. Congratulations for your work! However, it is necessary to improve the manuscript at some points before the publication:
- In general, I recommend an English review, as some sentences are incorrectly structured and some expressions are literally traduced from Italian.
AUTHORS’ RESPONSE: A NATIVE ENGLISH SPEAKER KINDLY TOOK THE TIME TO REVISE OUR MS. SEE ACKNOWLEDGEMENTS.
- Please pay attention at the page numbering. It starts 1 to 4 and after table 1, it starts again 1 to 7....
AUTHORS’ RESPONSE: OK, CORRECTED ACCORDINGLY.
- L32. You repeat twice the verb to sample in the same sentence. Please, use a synonym for one of them. This sentence is hard to read and understand, please reword it.
AUTHORS’ RESPONSE: OK, REWORDED. LINES 32-33.
- L38. Put Chalcides ocellatus in brakets, please.
AUTHORS’ RESPONSE: OK, DONE. LINE 39.
- L39. Please change thr to the.
AUTHORS’ RESPONSE: THE ARTICLE “THE” HAS BEEN DELETED.
- L52. "transport goods"... I think this expression is not well traduced...
AUTHORS’ RESPONSE: WE CHANGED IT IN “TRADE OF GOODS”. LINE 53.
- L56. Italy includes more municipalities than only coastal ones. Please reformulate the sentence.
AUTHORS’ RESPONSE: WE REPHRASED THE SENTENCE ACCORDINGLY. LINES 57-59.
- L62. Please change "If on one hand" to "In spite of" or "Despite".
AUTHORS’ RESPONSE: WE CHANGED IT WITH “ALTHOUGH”. LINE 63.
- L63-64. Please reformulate "mainly because of single released individuals"
AUTHORS’ RESPONSE: WE CHANGED IT IN “AS REGARDING ONLY SINGLE INDIVIDUALS”. LINES 64-65.
- L67. Please, put Mediodactylus kotschyix in brakets.
AUTHORS’ RESPONSE: OK, DONE. LINE 67.
- L89. "mainly dealt with" doesn't make sense. Please reformulate.
AUTHORS’ RESPONSE: WE CHANGED “MAINLY DEALT WITH” WITH “INCLUDED”. LINE 90.
- L99. Please change "cultivated area" to "crop area"
AUTHORS’ RESPONSE: OK CHANGED. LINE 100.
- L111. Please add the year of sample collection, not only the months.
AUTHORS’ RESPONSE: OK DONE. LINE 113.
- L121. I don't find in the text the meaning of GM, FPF, MDN and SM. For the first time you use an abbreviation in a scientific text it is important to put the meaning. For example: ... the pulse-field gel electrophoresis (PFGE)... Please do the same in L162 and 163 with HKY and GTR respectively.
AUTHORS’ RESPONSE: THESE ARE THE INITIALS OF SOME AUTHORS. NOW WE WROTE THE NAMES IN EXTENSO AND WE SHOULD HAVE CLARIFIED. LINES 122-124; 166-168.
- L143. "...were the same reported by (19)...". In order to make easier the reading, please change this to "...were the same reported by Kornilios et al. (2010) (19). Please do the same in L144, 262 and 263 with references 30 and 23 respectively.
AUTHORS’ RESPONSE: OK DONE, LINES 147, 148, 284.
- L150. "the following steps were followed". Please reformulate...
AUTHORS’ RESPONSE: OK REFORMULATED ACCORDINGLY. LINE 155.
- Figure 3 and 4. It results very difficult to read because of the small print. Please try to enlarge the text...
AUTHORS’ RESPONSE: WE ENLARGED THE TEXT IN BOTH FIGURES.
- Table 3. The title is incorrect.
AUTHORS’ RESPONSE: OK WE CORRECTED THE TITLE OF THE TABLE ACCORDINGLY. LINES 222-224.
- L253. Please put Fibla maclachlani in brackets.
AUTHORS’ RESPONSE: OK, DONE. LINE 270.
- L272-273. You repeat Therefore twice. Please change one of them...
AUTHORS’ RESPONSE: WE DELETED ONE “THEREFORE”.
Reviewer 2
Although the work presents relevant information for the study of invasive populations of a lizard species in Italy, this contribution seems too simple to be presented in a full article and seems more suitable for a ´short communication´.
AUTHORS’ RESPONSE: WE ARE AWARE THAT OUR SAMPLE COULD BE LIMITED, HOWEVER WE ALSO DESCRIBED RANGE AND METHODS IN A NUMBER OF WORDS WHICH DOES NOT ALLOW TO BE REDUCED FOR A SHORT COMMUNICATION. THUS, WE WOULD PREFER TO KEEP OUR MS AS A FULL PAPER. IF THE EDITOR WOULD ASK US TO MAKE IT A SHORT COMMUNICATION, WE WOULD THINK MORE ABOUT THIS.
In addition, the work presents some issues that weaken the quality of the manuscript. First, it is based on limited genetic data (few mitochondrial DNA-only loci) from a limited number of individual samples. Such limitations should at least be discussed in the work, but they are deliberately ignored.
AUTHORS’ RESPONSE: WE CLARIFIED THE LIMITATIONS OF OUR WORK IN THE DISCUSSION. LINES 243-247.
Often the results presented in the text do not appear clearly in the figures and tables. Figures are confusing and with low resolution. I couldn't even check the reliability of the phylogenetic trees, as the posterior probability values were unreadable. It seems like the article was not prepared with due care.
AUTHORS’ RESPONSE: WE IMPROVED THE QUALITY OF ALL OUR FIGURES.
The work claims in the 'simple summary' to present an unprecedented record of the species in central Italy, but does not present this information clearly in the results and does not discuss the relevance of this record. How much does this extend from the species' previous distribution? What does this represent for the problem of biological invasion by the species? In fact, the discussion and conclusion have little to do with the results of the work.
AUTHORS’ RESPONSE: WE THANK THE REVIEWER FOR THE SUGGESTION. NOW, WE HAVE ADDED A PART AT THE START OF THE RESULTS AND DISCUSSION INCLUDING THE DETECTION OF THE NEW POPULATION PREVIOUSLY UNRECORDED. WE ALSO IMPROVED OUR DISCUSSION IN TERMS OF THE IMPORTANCE OF MONITORING ALIEN SPECIES AND POTENTIAL IMPACTS. LINES 182-186; 228-233; 276-280.
Finally, I'm not sure it's worth keeping up with the two phylogenetic analyzes, as comparisons between their results undermine the study's conclusions. Perhaps keeping only the analysis with the largest number of samples is a more interesting option.
AUTHORS’ RESPONSE: WE UNDERSTAND THE POINT OF THE REVIEWER. IN OUR OPINION, IT WOULD BE BETTER TO INCLUDE BOTH PHYLOGENETIC RECONSTRUCTIONS, AS THE SECOND ONE – DESPITE INCLUDING A LOWER NUMBER OF SAMPLES – HAS BEEN BUILT UP WITH THREE GENES INCLUDING THOSE COMMONLY USED FOR BARCODING ANALYSES. WE IMPROVED THIS PART IN THE DISCUSSION. LINES 189-190; 244-247.
Minor comments:
Line 39: typo ´thr´
AUTHORS’ RESPONSE: THE ARTICLE “THE” HAS BEEN DELETED.
Line 47: typo ´Reptilila´
AUTHORS’ RESPONSE: OK, CORRECTED ACCORDINGLY. LINE 48.
Line 132: typo ´Chalcided´
AUTHORS’ RESPONSE: OK, CORRECTED ACCORDINGLY. LINE 135.
Figure 2: The white patches inside gray areas (islands) are confusing. I suppose that these patches represent areas where the species are absent. But without further explanation it is intriguing why these lizard-free areas are so specific and well delimited. I believe that it would be more accurate to inform only the localities - through numbered points on the map - where the species has already been effectively registered.
AUTHORS’ RESPONSE: THE REVIEWER IS RIGHT, NOW WE HAVE IMPROVED THE MAP RESOLUTION AND CLARIFIED THAT THE DISTRIBUTION OF THE OCELLATED SKINK IS REPRESENED BY GREY AREAS. WE THINK THAT, IF THE REVIEWER IS OK WITH OUR IDEA, IT WOULD BE BETTER TO KEEP THE GREY AREA TO SHOW THE ACTUAL DISTRIBUTION OF THE SPECIES IN ITALY, ALSO SHOWING THAT, DESPITE HAVING FEW SAMPLES (AS THE REV.2 HERSELF/HIMSELF HIGHLIGHTED), ALMOST ALL AREAS WITH THIS SPECIES INCLUDED AT LEAST ONE SAMPLE IN OUR ANALYSIS. LINES 108-111.
Figures 3 and 4 are too small. Please put them in a size and resolution good enough to make all the important information readable. The two figures can also be more informative. I strongly suggest, for instance, highlighting the two main clades referred to in the text.
AUTHORS’ RESPONSE: WE IMPROVED THE RESOLUTION OF OUR IMAGES AND HIGHLIGHTED CLADES AS SUGGESTED.
Tables 1 and 2: Please include the species´ name in the tables´ titles.
AUTHORS’ RESPONSE: OK, DONE IN ALL TABLE CAPTIONS. LINES 128, 144 AND 222-223.
Table 3: Wrong title (copy/paste from table 2…)
AUTHORS’ RESPONSE: OK WE CORRECTED THE TITLE OF THE TABLE ACCORDINGLY. LINES 222-223.
Reviewer 3
This article describes the discovery of some individuals of Chalcides ocellatus introduced in mainland Italy, likely as a result of maritime cork trade. The description is accompanied by some phylogenetic analyses that attempt to explain the potential origin of these individuals. The text is well written and easy to follow. Besides some minor grammar errors and some clarifications that need to be done, I do not have much to object to it.
AUTHORS’ RESPONSE: MANY THANKS INDEED.
Line 32: this is a bit redundant. Delete “and we sampled them”.
AUTHORS’ RESPONSE: THE SENTENCE HAS BEEN REPHRASED. LINES 32-33
Lines 32-34: the reader could use knowing now whether these populations are natural or introduced. The simple summary must be understandable without the aid of the rest of the text.
AUTHORS’ RESPONSE: WE HAVE NOW CLARIFIED THE SIMPLE SUMMARY ACCORDINGLY. LINES 31-34.
Line 39: did “thr” mean “the”?
AUTHORS’ RESPONSE: THE ARTICLE “THE” HAS BEEN DELETED.
Line 57: of the human population in the country, right? Make it clear.
AUTHORS’ RESPONSE: OK, CLARIFIED ACCORDINGLY. LINE 58.
Line 62: delete “the”.
AUTHORS’ RESPONSE: OK, DELETED ACCORDINGLY.
Line 62: “on the one hand”.
AUTHORS’ RESPONSE: OK, WE CHANGED IN “ALTHOUGH”. LINE 63.
Line 63: rather than “build”, I’d use “establish” here.
AUTHORS’ RESPONSE: OK, CHANGED ACCORDINGLY. LINE 64.
Line 93: delete “about”.
AUTHORS’ RESPONSE: OK, DELETED ACCORDINGLY.
Lines 111-122: it would be nice to see the sample sizes for each location clearly stated here.
AUTHORS’ RESPONSE: ONE SAMPLE WAS ANALYSED FROM EACH PLACE, AS STATED IN THE CAPTION. LINE 111.
Line 116: “particularly”.
AUTHORS’ RESPONSE: OK, CORRECTED ACCORDINGLY. LINE 118.
Line 132: “Chalcides”. Also, I’d say “samples” instead of “specimens”.
AUTHORS’ RESPONSE: OK, CORRECTED ACCORDINGLY. LINE 135.
Line 133: Please, briefly state what those outgroups were.
AUTHORS’ RESPONSE: WE ADDED THIS INFORMATION. LINE 136.
Line 134: “was extracted”.
AUTHORS’ RESPONSE: WE ADDED “WAS”. LINE 137.
Line 247: “suggests”.
AUTHORS’ RESPONSE: OK, CORRECTED. LINE 264.
Line 258: “different Sardinian” what?
AUTHORS’ RESPONSE: THAT WAS A MISPRINT, WE MEANT “SEVERAL SPECIES”. NOW, WE CORRECTED. LINE 275.
Line 262: “thus confirming” what? Did you mean “thus confirming the conclusions in [23]”?
AUTHORS’ RESPONSE: YES, WE HAVE NOW CLARIFIED. LINES 283-284.
Line 268: I don’t think this reference matches the journal style.
AUTHORS’ RESPONSE: YES, THIS WAS A MISPRINT, WHICH HAS BEEN NOW CORRECTED. LINE 290.
Reviewer 2 Report
Although the work presents relevant information for the study of invasive populations of a lizard species in Italy, this contribution seems too simple to be presented in a full article and seems more suitable for a ´short communication´. In addition, the work presents some issues that weaken the quality of the manuscript. First, it is based on limited genetic data (few mitochondrial DNA-only loci) from a limited number of individual samples. Such limitations should at least be discussed in the work, but they are deliberately ignored.
Often the results presented in the text do not appear clearly in the figures and tables. Figures are confusing and with low resolution. I couldn't even check the reliability of the phylogenetic trees, as the posterior probability values were unreadable. It seems like the article was not prepared with due care.
The work claims in the 'simple summary' to present an unprecedented record of the species in central Italy, but does not present this information clearly in the results and does not discuss the relevance of this record. How much does this extend from the species' previous distribution? What does this represent for the problem of biological invasion by the species? In fact, the discussion and conclusion have little to do with the results of the work.
Finally, I'm not sure it's worth keeping up with the two phylogenetic analyzes, as comparisons between their results undermine the study's conclusions. Perhaps keeping only the analysis with the largest number of samples is a more interesting option.
Minor comments:
Line 39: typo ´thr´
Line 47: typo ´Reptilila´
Line 132: typo ´Chalcided´
Figure 2: The white patches inside gray areas (islands) are confusing. I suppose that these patches represent areas where the species are absent. But without further explanation it is intriguing why these lizard-free areas are so specific and well delimited. I believe that it would be more accurate to inform only the localities - through numbered points on the map - where the species has already been effectively registered.
Figures 3 and 4 are too small. Please put them in a size and resolution good enough to make all the important information readable. The two figures can also be more informative. I strongly suggest, for instance, highlighting the two main clades referred to in the text.
Tables 1 and 2: Please include the species´ name in the tables´ titles.
Table 3: Wrong title (copy/paste from table 2…)
Author Response
Here our replies to Reviewer 2

Reviewer 3 Report
This article describes the discovery of some individuals of Chalcides ocellatus introduced in mainland Italy, likely as a result of maritime cork trade. The description is accompanied by some phylogenetic analyses that attempt to explain the potential origin of these individuals. The text is well written and easy to follow. Besides some minor grammar errors and some clarifications that need to be done, I do not have much to object to it.
Line 32: this is a bit redundant. Delete “and we sampled them”.
Lines 32-34: the reader could use knowing now whether these populations are natural or introduced. The simple summary must be understandable without the aid of the rest of the text.
Line 39: did “thr” mean “the”?
Line 57: of the human population in the country, right? Make it clear.
Line 62: delete “the”.
Line 62: “on the one hand”.
Line 63: rather than “build”, I’d use “establish” here.
Line 93: delete “about”.
Lines 111-122: it would be nice to see the sample sizes for each location clearly stated here.
Line 116: “particularly”.
Line 132: “Chalcides”. Also, I’d say “samples” instead of “specimens”.
Line 133: Please, briefly state what those outgroups were.
Line 134: “was extracted”.
Line 247: “suggests”.
Line 258: “different Sardinian” what?
Line 262: “thus confirming” what? Did you mean “thus confirming the conclusions in [23]”?
Line 268: I don’t think this reference matches the journal style.
Author Response
Here our replies to reviewer 3

Round 2
Reviewer 2 Report
The presentation of the manuscript has been considerably improved. However, I still think that the relevance of the results presented would fit better in a short communication. The authors claim that the description of the methods is too long for that, but I don't think this is a reasonable argument for defining what should or should not become a complete article. However, I agree to leave the final decision on this to the editor.